

# Exploring effects of variation in plant root traits on carbon emissions from estuarine marshes

Youssef Saadaoui[1], Christian Beer[2,3], Peter Mueller[4], Friederike Neiske[2,3], Joscha N. Becker[2,3], Annette Eschenbach[2,3], Philipp Porada[1,3]

[1]Universität Hamburg; Faculty of Mathematics, Informatics and Natural Sciences; Department of Biology; Institute of Plant Science and Microbiology, Ohnhorststr. 18, 22609 Hamburg, Germany
[2]Universität Hamburg, Faculty of Mathematics, Informatics and Natural Sciences, Department of Earth System Sciences; Institute of Soil Science, Allende-Platz 2, 20146 Hamburg, Germany
[3]Universität Hamburg, Center for Earth System Research and Sustainability, Bundesstraße 53, 20146 Hamburg, Germany
[4]University of Münster, Institute of Landscape Ecology, Heisenbergstr. 2, 48149 Münster, Germany

*Correspondence to*: Youssef Saadaoui (youssef.saadaoui@uni-hamburg.de)

**Abstract.** Estuarine marshes are crucial components of coastal environments around the world and provide numerous ecosystem services, such as carbon sequestration. Plant-microbe interactions are potential key drivers of organic carbon cycling in these ecosystems, but their contribution to the ecosystem-level carbon balance has been rarely quantified so far. This is partly due to the substantial intra- and interspecific variation of plant traits that are affecting microbial functions. Traits such as root oxygen loss and root exudation, for instance, modify soil heterotrophic respiration, but may strongly differ between plant species. Moreover, the non-linearity of the relationships between soil carbon fluxes and effects of plant-microbe interactions may require an explicit representation of trait variation for correctly estimating the carbon balance of estuarine marshes in ecosystem models. However, modelling approaches in this regard so far mostly represent plants as a set of traits that are based on average values of different individuals or species, thus not capturing trait variation. In this study, we implemented a key plant trait, the modification of soil oxygen concentration, into a simple model of heterotrophic respiration in estuarine marsh soils. We then compared two model configurations, one with and one without explicit representation of variation in soil oxygen levels, to estimate the effect on simulated heterotrophic respiration. We found a 10% reduction in the average respiration rate and a deviation from the median of +33% /-47% within the first and third quartile of the distribution in the approach that accounted for trait variation. This illustrates the potentially large impacts that may arise from spatial heterogeneity of plant species or changing community composition of plants on the carbon balance of estuarine marshes. We thus suggest implementing trait variation in marsh ecosystem models.

## 1 Introduction

Estuarine marshes are ecosystems at the intersection of land and sea that are distinguished by their salt-tolerant plant species, diverse microorganisms, tidal dynamics, and variation in salinity. These marshes play an essential role for the global coastal environment with regard to ecosystem functions (Kirwan et al., 2016a) and they offer a variety of ecosystem services such as



shoreline protection and water purification, i.e. filtering pollutants and excess nutrients before they reach coastal waters
(Barbier et al., 2011). Estuarine marshes are crucial for carbon sequestration as they can store large amounts of carbon in their
soil, mostly due to trapping of carbon-rich sediments and high primary productivity combined with slow decomposition of
organic matter (Granse et al., 2024). The marshes are also important for nutrient cycling and as a habitat for a wide range of
species (Duarte et al., 2013).

Plant-microbe interactions are key drivers of carbon cycling in estuarine marshes, as they affect both primary production and
soil organic matter decomposition. One crucial process in this regard is root oxygen loss (ROL) by plants, which may create
oxygen-rich microenvironments in the soil, thereby affecting microbial activity and pathways and, consequently,
decomposition rates (Mueller et al., 2016). Spatial variation in oxygen levels plays a substantial role for the carbon balance of
estuarine marsh ecosystem (Spivak et al., 2019), since high oxygen levels promote aerobic decomposition and large emissions
of carbon in the form of $CO_2$, while low oxygen availability results in anaerobic processes that cause small release of carbon
as methane ($CH_4$) (Bhattacharyya and Furtak, 2022). Soil microbes, in turn, affect plants by releasing nutrients during
decomposition, thus influencing plant productivity and ecosystem carbon input (Wang et al., 2024). Hence, understanding
plant-microbe interactions and their impacts on soil oxygen dynamics is essential for quantifying the carbon balance of
estuarine marshes and predicting their response to environmental changes such as sea-level rise and climate change (Crump
and Bowen, 2024; Kirwan and Blum, 2011; Spivak et al., 2023; Tang et al., 2021; Weingarten et al., 2023).


In recent years, process-based modelling approaches have been used to examine which biotic and abiotic factors drive the
carbon balance of estuarine marshes. These models usually simulate processes such as photosynthesis, respiration, and nutrient
cycling to estimate carbon fluxes, and represent plants by several plant functional types (PFTs), while interactions with
microbes are often only implicitly considered. The Wetland-DNDC Model, for instance, assumes that soil microbial processes
associated with various biogeochemical cycles in wetland ecosystems are driven mostly by plant litter, soil temperature, and
hydrology (Zhang et al., 2002). Subsequent models have refined the representation of marsh ecosystem processes (Morris et
al., 2002; Swanson et al., 2014), but concentrated mainly on linking marsh elevation changes and vegetation productivity to
simulate carbon sequestration, not accounting for ROL and root exudation. Several further models (Alizad et al., 2016; Kirwan
et al., 2016b) simulate the feedbacks between plant biomass and geomorphological evolution of marsh surfaces and channels
with increasing levels of complexity, but do not consider explicit plant-microbe interactions. More recently, however, (Rietl
et al., 2021) focused on how vegetation type and priming of decomposition by plants affects carbon accumulation in brackish
marshes under global change.

While partly considering some aspects of plant-microbe interactions, the current marsh ecosystem modelling approaches
usually aggregate the functional diversity of marsh plants into a few functional types or representative species that are
characterized by averaged trait values (e.g. (Rietl et al., 2021; Swanson et al., 2014)). This means that key plant traits that





affect soil microbial carbon cycling, such as ROL or root exudation, for instance, do not show substantial variation in the models, if considered in the first place. In reality, however, plants strongly differ in these traits both at the intra- and the interspecific level. Studies have shown that there is significant intraspecific variation in ROL (Koop-Jakobsen et al. 2021) and

soil microbial carbon cycling among different genotypes of the same plant species (Tang et al., 2021). Additionally, interspecific differences are notable, with some species demonstrating higher root oxygen loss compared to others (Bernal et al., 2017; Mueller et al., 2020; Van Der Nat and Middelburg, 1998). Root oxygen loss can vary widely between species, with reported values ranging from 5 to 50 mmol $O_2$ $m^{-2}$ $day^{-1}$ depending on the species (Visser et al., 2000). As a consequence, microbial communities in marsh soils may experience substantial variation in resource availability at a small scale and will

thus differ largely in their local rates of respiration. This complicates the model-based estimation of soil carbon emissions at the ecosystem level. Furthermore, the relationships between heterotrophic respiration rates and amounts of soil carbon and oxygen are non-linear. This means, for instance, that the average respiration rate derived from a range of different oxygen levels will differ from the rate derived from one average oxygen level, which is known as aggregation error (Rastetter et al., 1992). Therefore, to accurately estimate the effects of plant-microbe interactions on the carbon balance of estuarine marshes,

it is essential to explicitly represent the spatial and temporal distribution of microbe-related trait values within plant communities. This includes considering both seasonal variations and long-term changes.

Process-based numerical vegetation models that explicitly represent functional diversity, i.e. trait ranges of plants, have been increasingly developed in the last few decades (e.g. (Butler et al., 2017; Pavlick et al., 2012; Scheiter et al., 2013; Snell et al.,

2014)). These models mainly aim at capturing the effects of (changing) climatic conditions on the community composition of plants, thereby focusing mostly on forests, grasslands and net primary productivity. To date, the consequences of variation in plant traits for soil microbial functions have been rarely studied with these models, and this approach has not been applied to estuarine marsh ecosystems so far (Piercy et al., 2024).

Here, we estimate the effect of plant-induced variation of oxygen levels in soils on heterotrophic respiration in estuarine marshes using the DAMM model (Davidson et al., 2012). To estimate the aggregation error, we simulate the respiration response to soil carbon for a range of different oxygen levels and compare this to a model configuration that only considers one average soil oxygen level.

## 2 Materials and methods

The Dual Arrhenius and Michaelis—Menten (DAMM) kinetics model has been developed to simulate soil organic matter decomposition at hourly to seasonal time scales. It combines two processes that have been shown to substantially affect decomposition rates: temperature dependence, represented by the Arrhenius model, and substrate vs. enzymatic limitation,





represented by the Michaelis-Menten model. The model was developed by (Davidson et al., 2012) to improve the representation of soil organic matter decomposition in Earth system models.


We selected the DAMM model to test the effect of variation in root oxygen loss on soil microbial activity and ecosystem carbon emissions since it includes an additional term for oxygen limitation of heterotrophic respiration (Eq. 1), while most models of soil organic matter decomposition focus only on carbon as a substrate, being developed for well-aerated soils.

To calculate heterotrophic respiration rate, $R_{sx}$, in the model we used Eq. (1):

$$R_{Sx} = V_{\max} \times \frac{[S_x]}{kM_{sx}+[S_x]} \times \frac{[O_2]}{kM_{o2}+[O_2]} , \tag{1}$$

Where $V_{\max}$ is the maximum reaction velocity when both substrates are not limiting. $[S_x]$ and $[O_2]$ denote the concentrations of soluble soil organic carbon and soil oxygen, respectively. The constants $kM_{Sx}$ and $kM_{O2}$ are the corresponding half-saturation constants. Vmax is calculated according to the Arrhenius function in Eq. (2):

$$V_{max} = \alpha_{Sx} \times e^{\frac{-Ea_{Sx}}{RT}} , \tag{2}$$

Where R is the Universal gas constant, and T is the temperature in Kelvin. $\alpha_{Sx}$ and $Ea_{Sx}$ are a pre-exponential factor and the activation energy of the reaction, respectively.

The concentration of soluble carbon $[S_x]$ is calculated using Eq. (3):

$$[Sx] = C_{org} \times p \times D_{liq} \times \theta^3 , \tag{3}$$

where $C_{org}$ is the total amount of the soil organic carbon, p is the soluble fraction, $D_{liq}$ is the diffusion coefficient for the liquid phase, and $\theta$ is the volumetric water content of the soil.

To represent root oxygen loss in the model, we use the dependence of soil oxygen concentration on soil moisture that is already

implemented in the model according to Eq. (4):

$$[O_2] = D_{gas} \times 0.209 \times a^{\frac{4}{3}} , \tag{4}$$

where $D_{gas}$ is the diffusion coefficient for $O_2$ in air, 0.209 is the volume fraction of $O_2$ in air, and a is the air-filled porosity of

the soil, which is calculated as follows Eq. (5):

$$a = 1 - \frac{BD}{PD} - \theta , \tag{5}$$





where BD is the bulk density, PD is the particle density, and θ is the soil volumetric water content. By variation of θ, we modify the diffusion rate of oxygen in the soil and thus mimic root oxygen loss. While this is not a mechanistic representation, the effect of plant aerenchyma on the diffusion of oxygen from the atmosphere into the soil is similar to the effect of increased

air-filled pore space during decreasing soil moisture. Since we do not simulate the impacts of soil moisture on plant or microbial physiological processes, our results are not affected by this approximation.

## 2.1 Model experiment

To quantify the effect of trait variation on microbial respiration, we compared two different model configurations:

(1) Respiration rate was simulated at constant temperature for a range of different soil organic carbon contents (0.001 to 0.3 g C / cm$^3$ soil), and the soil oxygen concentration was assumed to remain at a constant, average value of 0.097 cm$^3$ O$_2$/ cm$^3$ air, corresponding to 30% relative soil moisture. This corresponds to representing in the model an average plant type with a fixed value of root oxygen loss.

(2) Same as setup (1), but we simulated the response of respiration to soil organic carbon content for 70 different concentrations

of soil oxygen and subsequently calculated the average respiration of all oxygen levels. To constrain the oxygen values, we used Eqs. 3 and 4 of the model and varied soil water content from the dry state to full saturation.

## 2.2 Parametrisation

To make our model setup consistent with carbon cycling in estuarine marshes, we adapted the parameters of the original

DAMM model, based on a series of incubation experiments carried out with soil samples from the Elbe marshes near Hamburg, Northern Germany. The samples were incubated under aerobic conditions for a period ranging from 316 to 465 days. As a first step, we calibrated the Introductory Carbon Balance Model (ICBM) (Andren and Katterer, 1997) to the incubation data to be able to estimate respiration rates for a larger range of soil organic carbon contents than were included in the laboratory incubation runs. Methodological details can be found in (Knoblauch et al., 2013) and (Beer et al., 2022). This was done since

the saturation effects of the DAMM model are only apparent at relatively high substrate levels. Subsequently, we used the respiration rates simulated after the first 30 days of the incubation from 300 ICBM runs that differed in initial soil organic carbon content (Fig. 1). This period corresponds roughly to average respiration rates under field conditions, where fresh organic matter is continuously provided, preventing a strong decline in respiration rates due to substrate limitation (Fig. 1).





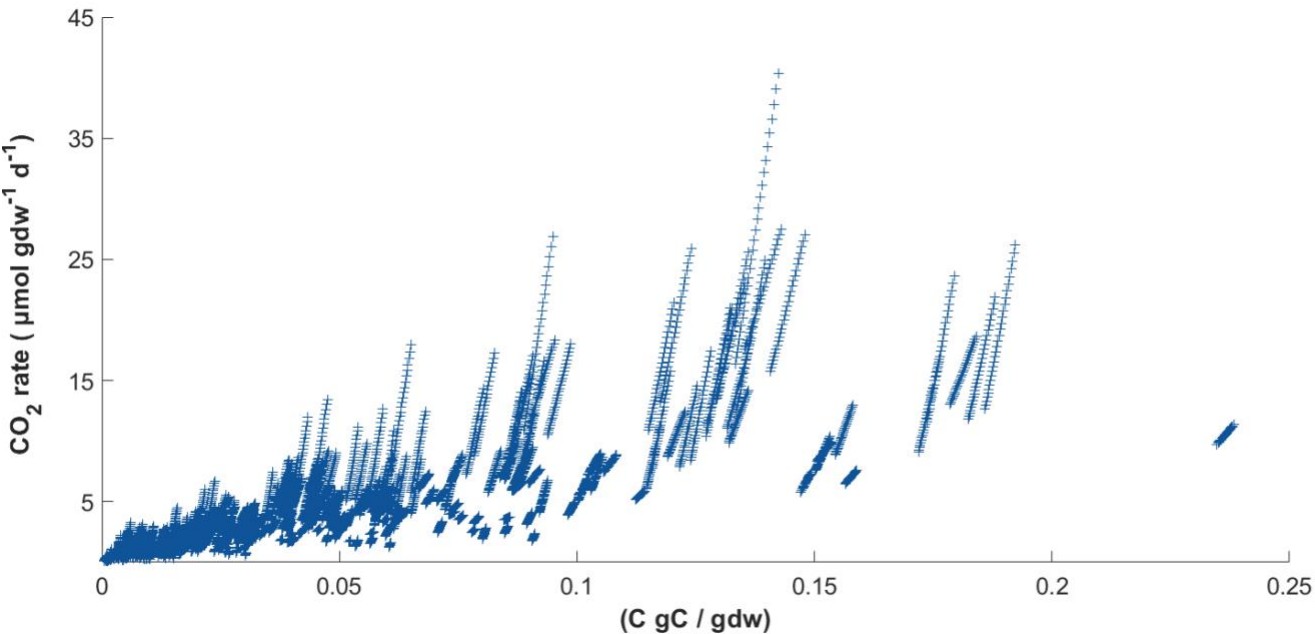

**Figure 1:** Respiration rates simulated by the ICBM during the first 30 days of incubation with varying initial soil organic carbon contents (gram carbon per gram dry weight). We assume that this period captures the fluctuations of respiration rates under field conditions where fresh organic matter is continuously provided. Oxygen limitation is not considered here as the incubation was carried out under aerobic conditions.

Finally, we computed median values of ICBM-simulated respiration for seven bins of the considered soil organic carbon content range and fitted the DAMM model to these data points by visual comparison (Fig. 2). To this end, we varied the parameters $\alpha_{Sx}$, (Eq. 2) and $kM_{Sx}$ (Eq. 1), thus altering $V_{max}$ and the substrate concentration at which the reaction nears saturation. Other parameters were the same as in the original DAMM model publication (Davidson et al., 2012). See Tab. 1 for the model parameters.



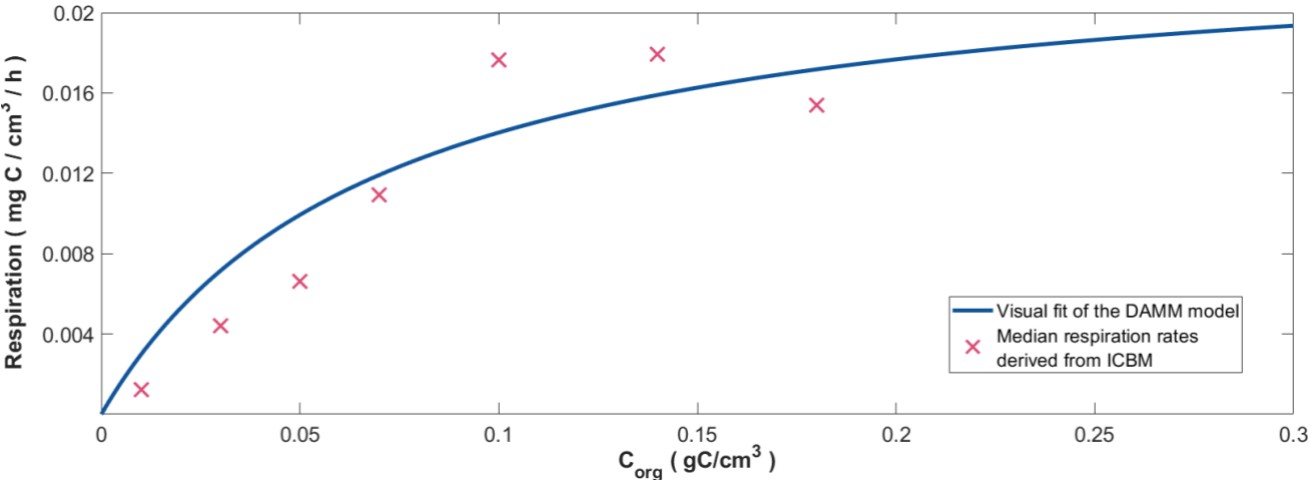


**Figure 2:** Visual fit of the DAMM model (blue line) to median respiration rates (magenta crosses) derived from ICBM simulations across varying soil organic carbon contents.

**Table 1: Model parameters:**

| Parameters | Value | Unit | Parameters | Value | Unit |
|---|---|---|---|---|---|
| $kM_{Sx}$ | 2.4875E-06 | $(gC/cm^3)$ | θ avg | 0.3 | $(cm^3\ H_2O/\ cm^3\ soil)$ |
| $kM_{O2}$ | 0.121 | $(cm^3\ O2/\ cm^3\ air)$ | Range of θ | [0 - 0.6825] | $(cm^3\ H_2O/\ cm^3\ soil)$ |
| $Ea_{Sx}$ | 72.26E+3 | $(J\ mol^{-1})$ | $D_{liq}$ | 3.17 | - |
| $\alpha_{Sx}$ | 4.0350E+11 | $(mgC/cm^3/h)$ | $D_{gas}$ | 1.67 | - |
| R | 8.314 | $(J\ mol^{-1}\ K^{-1})$ | BD | 0.8 | $(g/cm^3)$ |
| T | 293.15 | (K) | PD | 2.52 | $(g/cm^3)$ |
| p | 4.14E-4 | - | | | |

## 3 Results and discussion:

By comparing the two model configurations (see Fig. 3) we find that the variation of the plant trait root oxygen loss leads to a 10% reduced respiration rate, when averaged over all soil oxygen levels, compared to the model configuration driven only by one average soil oxygen concentration, corresponding to a model with one plant functional type. The percentage reduction is
averaged over the range of simulated soil carbon content. The first and third quartiles of the distribution of respiration responses comprise a deviation of +33% and -47% from the median curve, respectively.





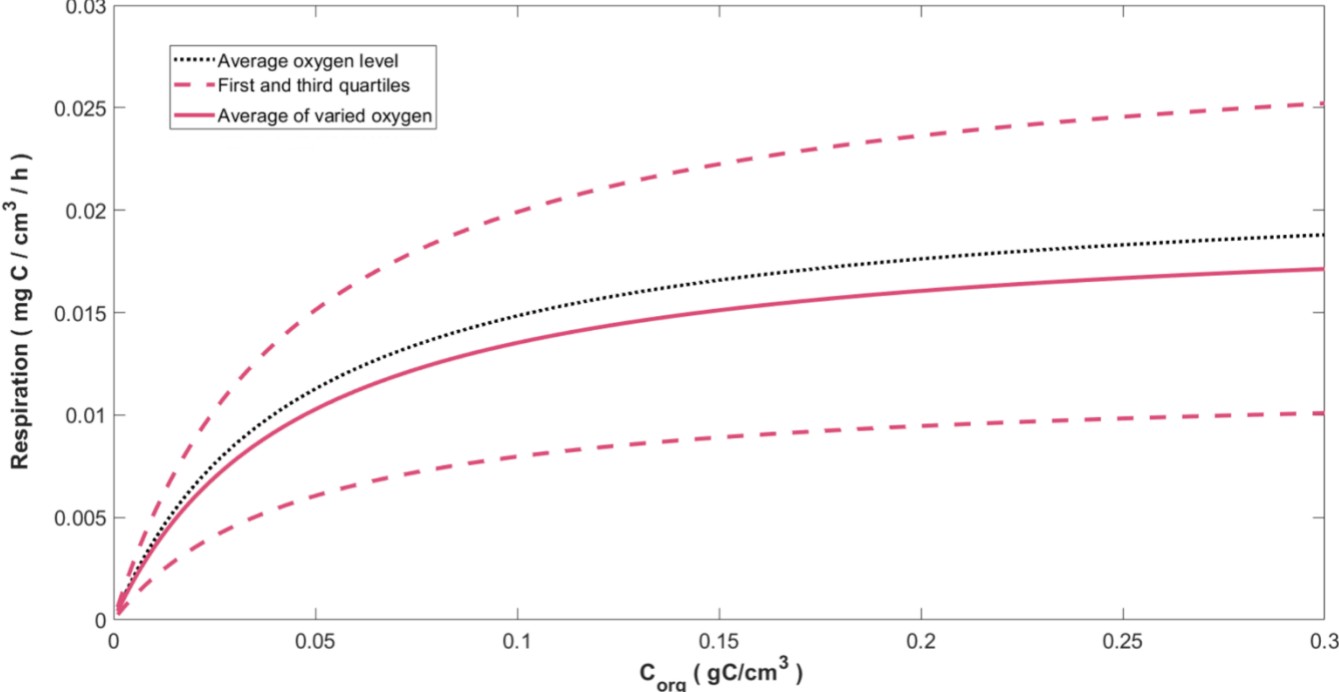

**Figure 3:** Comparison of heterotrophic respiration rates assuming an average (black) versus varied (magenta) root oxygen loss. The solid magenta line represents the average heterotrophic respiration response to soil carbon content across the entire range of oxygen
levels, while the red dashed lines denote the first and third quartiles of this range. The black dotted line indicates the simulated respiration response assuming an average soil oxygen concentration.

Our simple model experiment illustrates the potentially substantial impact of variation of plant traits associated with root oxygen loss and microbial respiration for the carbon balance of estuarine marshes. We considered a large range of possible
soil oxygen levels in the model. However, also under natural conditions, strong variation in oxygen may occur in the rhizosphere of marsh ecosystems, due to (I) tidal dynamics and associated water table fluctuations (II) increased respiration rates causing short-term oxygen deficiency at small scale or (III) vegetation dynamics that may lead to the occurrence /disappearance of species that have a high rate of root oxygen loss. It should be noted here that we assume a uniform frequency distribution of all possible oxygen levels at a given site in our modelling approach. To develop this further, a representation of
all relevant abiotic and biotic factors that cause variation in soil oxygen concentration would be required in marsh ecosystem models. In particular, different plant types /species, each characterized by microbe-related traits, and the (dynamic) abundance of these plants would have to be considered, but also dynamic soil hydrological and thermal conditions. Such a modelling

approach would not only be adequate to estimate the role of plant-microbe interactions for the current carbon balance of marsh ecosystems, but it would also allow more realistic predictions of carbon fluxes under climate change pressures, such as

warming and sea level rise. Considering only our simple approximation of varied root oxygen loss, and assuming global soil carbon emissions of 31 Tg a$^{-1}$ via heterotrophic respiration from salt marshes (Alongi, 2020), the estimated reduction by 10% would correspond to roughly 8% of global annual anthropogenic carbon emissions (Intergovernmental Panel on Climate Change (IPCC), 2023).

## Code availability

The MATLAB codes are available upon request from the corresponding author and will be made available for download from a repository.

## Data availability

The data is available upon request from the corresponding author. The ICBM data can be downloaded from a repository; access will be granted upon request. http://doi.org/10.25592/uhhfdm.14367.

## Author contribution

YS, PP, CB and PM designed the study and YS, PP, and CB carried out the modelling. FN, JNB and AE designed the incubation experiment, FN investigated and provided the incubation data. YS wrote the manuscript with input from all co-authors.

## Competing interests

The authors declare that they have no conflict of interest.

## Acknowledgements

This project was funded by the Deutsche Forschungsgemeinschaft (DFG, German Research Foundation) as a part of the research training group (RTG2530) "Biota-mediated effects on Carbon Cycling in Estuaries (grant no. 407270017).

P.M was supported by Deutsche Forschungsgemeinschaft (DFG) in the framework of the Emmy Noether program (502681570).

C.B. acknowledges financial support by the Heisenberg professorship (DFG-BE 6485/4-1).

We thank Daniel Schwarze for supporting laboratory work.



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
