# Peer review of "Exploring effects of variation in plant root traits on carbon emissions from estuarine marshes"

_EGUsphere, 2024_

## Author Comment (AC2)

**Response to Reviewer #2**

We thank Reviewer #2 for their thorough evaluation and constructive feedback, which has helped us significantly enhance the clarity, methodology, and framing of our manuscript. Below, we reproduce each of the reviewer's comments in italic text, followed by our detailed responses.

**General assessment, scope and novelty**

*"This study aimed to examine the effects of plant root traits on soil carbon emissions in estuarine marshes using a modeling approach. The authors conducted two model experiments using the DAMM model to simulate microbial respiration: one with constant soil O2 (fixed soil moisture) and another with variable soil O2 (influenced by soil moisture). They observed differing respiration response to soil carbon content under the two conditions.*

*While soil heterotrophic respiration in estuarine marshes is a critical component of carbon cycling and warrants further investigation, the study employed a relatively simple model based on incubation data. This limits the ability to extrapolate the findings across broader spatial and temporal scales. Additionally, the results are not particularly novel, as the role of O2 in regulating respiration under wetland conditions is well established. The link between the root traits and the model experiments is also unclear. The simulations primarily focused on the effects of soil moisture (and associated O2 availability) on respiration, rather than explicitly root traits. The authors should clarify their methodological framework and more clearly articulate the study's novelty."*

**Response:**

We thank the reviewer for these important points and have made explicit clarifications regarding the study's novelty and methodological framework. The novelty of our study lies specifically in quantifying the aggregation error that arises from spatial variability in rhizosphere oxygen ($O_2$) levels driven by root oxygen loss (ROL). This factor has not been quantified so far for estuarine marsh ecosystems, but is key for modelling their carbon fluxes. Below, we further elaborate on these points:

**Limited ability for extrapolation:**
Our objective was not to directly provide a landscape-scale parameterization. Instead, we explicitly aimed to assess the ecological relevance of $O_2$ heterogeneity driven by ROL for heterotrophic respiration in marsh ecosystems. Incubation experiments using real marsh soil samples are thus methodologically appropriate and robust. This intent is explicitly clarified in the manuscript:

"Here, we estimate the effect of plant-induced variation of oxygen levels in soils on heterotrophic respiration in estuarine
marshes using the DAMM model (Davidson et al., 2012). To estimate the aggregation error, we simulate the respiration
response to soil carbon for a range of different oxygen levels and compare this to a model configuration that only considers
one average soil oxygen level." (p. 3, L 90-93).

**Novelty regarding the role of $O_2$:**

We acknowledge that the general role of $O_2$ in wetland soil respiration is well-known. However, our study specifically addresses the quantitative implications of $O_2$ variation on respiration, highlighting the aggregation error introduced when spatial variation is replaced by a single mean $O_2$ value. This quantification has important implications for improving accuracy in large-scale ecosystem models. We have explicitly articulated this objective in the revised manuscript:

"By comparing the two model configurations (see Fig. 3) we find that the variation of the plant trait root oxygen loss leads to a 10% reduced respiration rate, when averaged over all soil oxygen levels, compared to the model configuration driven only by one average soil oxygen concentration... (p. 7- L 72-74).

"This illustrates the potentially large impacts that may arise from spatial heterogeneity of plant species or changing community composition of plants on the carbon balance of estuarine marshes." (p. 1- L 26-27).

"Considering only our simple approximation of varied root oxygen loss, and assuming global soil carbon emissions of 31 Tg $a^{-1}$ via heterotrophic respiration from salt marshes (Alongi, 2020), the estimated reduction by 10% would correspond to roughly 8% of global annual anthropogenic carbon emissions (Intergovernmental Panel on Climate Change (IPCC), 2023)." (p. 9- L 195-198)

A comparison with the detailed pore-scale model by Zhou et al. (2024) has also been included:

Additionally, we include a direct comparison with the detailed pore-scale model by Zhou et al. (2024):

"While Zhou et al. (2024) employed a highly detailed pore-scale model explicitly simulating plant-mediated oxygen transport and fine-scale biogeochemical processes, our approach quantifies aggregation errors resulting from spatial averaging in simpler ecosystem models. Thus, these approaches are complementary, each providing valuable insights into different aspects of oxygen dynamics and carbon cycling in estuarine marshes."

**Unclear link between root traits and experiment:**

The experiment was designed explicitly to estimate the effects of $O_2$ variation on respiration. Root traits serve as the underlying cause of this variation. We acknowledge that we do not explicitly simulate the mechanistic link between specific root traits and soil $O_2$ levels, but rather we impose observed $O_2$ variation derived from the literature. This approach allows us to isolate and quantify the aggregation error without introducing speculative parameters. We detailed this approach clearly in the revised manuscript:

"ROL itself is not simulated. Instead, its observed outcome, rhizosphere $O_2$ concentration, is imposed directly, following the empirical range detailed above. This allows us to quantify the impact of ignoring heterogeneity without speculative parameterisation of the many plant traits that underlie ROL." (P.6- L 227-229).

**Specific comments**

**1 Abstract**

*"Abstract: The background is too long, compared to the methods and results of this study. I recommend the authors to shorten the background information and expand the method part.."*

**Response:** We thank the reviewer for this suggestion. In the revised Abstract we shortened the background to two sentences and added one sentence that summarises the two simulation experiments.

Inserted text (p. 1, L 14-24):
" Estuarine marshes are crucial components of coastal environments around the world and provide numerous ecosystem services, such as carbon sequestration. Plant-microbe interactions are potential key drivers of organic carbon cycling in these ecosystems. Roots of marsh plants, for instance, leak oxygen into otherwise anoxic sediments, creating a patchy rhizosphere $O_2$ field that controls heterotrophic respiration. Thereby, traits such as root oxygen loss (ROL) may strongly differ between plant species, leading to strong variation in respiration rates. Ecosystem models that aim to quantify the carbon balance of marshes, however, usually replace this variety with one bulk value, leading to potential biases in carbon flux estimates. We calibrated a Dual Arrhenius–Michaelis–Menten (DAMM) scheme with laboratory incubations and ran two experiments: (i) uniform $O_2$ versus (ii) the full observed $O_2$ distribution. Spatial averaging of $O_2$ reduced the simulated aerobic $CO_2$ efflux by $\approx 10\,\%$. This illustrates the potentially large impacts that may arise from spatial heterogeneity of plant species or changing community composition of plants on the carbon balance of estuarine marshes. We thus suggest implementing trait variation in marsh ecosystem models."

**2 L21 – definition of the "root-trait"**

*"how can the modification of soil O2 concentration be considered as a key plant trait?"*

**Response** We thank the reviewer for raising this important conceptual point. In the revised manuscript we now clarify that the plant trait in question is the rate of root oxygen loss (ROL) rather than the resulting soil $O_2$ concentration itself. Following the generic trait definition of Violle et al. (2007) "any phenotypic property that affects ecosystem processes" ROL qualifies because it is an emergent, plant-controlled flux governed by root porosity, aerenchyma development, and metabolic demand. Numerous rhizosphere studies show that variation in ROL is the primary proximate control on local $O_2$ tension and hence on aerobic decomposition rates in marsh sediments (Visser et al., 2000; Armstrong & Armstrong, 2005; Kühl et al., 2021).

Inserted text (p. 3, L 149-152)

"In the functional-trait framework (Violle et al., 2007) we treat the the flux density of root oxygen loss (ROL) expressed as $O_2$ flux per root surface area as the focal trait, because ROL directly affetcs rhizosphere $O_2$ level and thereby regulates aerobic heterotrophic respiration (Visser et al., 2000; Armstrong & Armstrong, 2005; Kühl et al., 2021). Spatial and temporal variability in ROL therefore propagates to variability in soil $O_2$ without invoking additional soil-physical drivers."

**3   L40 – additional literature**

*"Please provide more references to link the plant roots to root O2 loss."*

**Response**   We added Holmer & Pedersen (2003) and the three papers cited above, expanding the reference base on intra- and interspecific variability in ROL.

Inserted text (p. 6, L 233-234)

"…intra- and inter-specific variation in ROL has been documented for marsh grasses, rice and sedges (Holmer & Pedersen, 2003; Visser et al., 2000; Armstrong & Armstrong, 2005; Kühl et al., 2021), supporting the empirical $O_2$ range used here."

**4   L90-91 – choice of the DAMM model**

*"Why the DAMM model was selected in this study?"*

**Response** We thank the reviewer for requesting a more detailed justification. The decisive argument is that DAMM contains an explicit Michaelis–Menten $O_2$-limitation term (Davidson et al., 2012). This mechanistic representation is essential for our objective of testing how different prescribed rhizosphere-$O_2$ distributions propagate into respiration.

The Dual-Arrhenius–Michaelis–Menten (DAMM) equation treats oxygen as a true saturating substrate ($K_s$-type). Consequently we can vary [O2] over several orders of magnitude— exactly the range observed around marsh roots while holding all other drivers constant. Most models of carbon cycling at the land-surface, by contrast, link respiration to soil moisture or redox potential, so $O_2$ cannot be manipulated independently and the aggregation error we target cannot be isolated. The Michaelis–Menten form also guarantees correct asymptotic behaviour, preventing the non-physical extremes that can arise from linear moisture scaling.

Inserted text (p. 4, L 163-170)

"We selected the DAMM model to test the effect of variation in root oxygen loss on soil microbial activity and ecosystem carbon emissions since it includes an additional term for oxygen limitation of heterotrophic respiration (Eq. 1), while most models of soil organic matter decomposition focus only on carbon as a substrate, being developed for well-aerated soils.

DAMM contains an explicit Michaelis–Menten $O_2$-limitation term (Davidson et al., 2012), allowing us to prescribe the full field-observed rhizosphere $O_2$ distribution and thereby quantify the resulting aggregation bias. Moreover, it requires only six parameters, which keeps the uncertainty analysis tractable. DAMM has also reproduced hourly $CO_2$ fluxes in tidal-freshwater marshes (Knox et al., 2021), demonstrating field applicability."

**5   L102 – model simplicity**

*"Many processed models considered the impacts of temperature, soil moisture (water-filled porosity), O2, and pH. But the model used here is a relatively simple one as showed below (Eq. 1)."*

**Response** We thank the reviewer for noting this. We now explain why a simple formulation is adequate for isolating the target bias.

Inserted text (p. 4, L 171-173):

Addition: "... We deliberately chose a simple modelling approach since the large parameter uncertainty at ecosystem or landscape scales would prevent an accurate estimate based on more complex models. Hence, there would be no reliable additional quantitative information that could be gained from exchanging the simple with a complex model."

Inserted text (p. 4, L 176-178):

"Redox-pH schemes for tidal soils require $> 40$ marsh-specific parameters; with so many unknowns, adding complexity would obscure rather than illuminate the aggregation error that we wish to quantify. A six-parameter DAMM therefore represents a parsimonious tool for the present goal, in our opinion."

**6  L 110 – merge Eqs 1 & 2**

*"Merge Eq. 2 into Eq. 1."*

**Response.** Implemented as requested. The merged equation now integrates temperature, substrate and $O_2$ limitation; all symbols are listed in Table 1.

Inserted text (p. 4, L 181-195)

"The heterotrophic respiration rate R (mg C cm$^{-3}$ h$^{-1}$) is given by:

$$R = \propto_{sx} e^{\left(\frac{-E_a}{RT}\right)} \times \frac{S_x}{K_{msx} + S_x} \times \frac{[O2]}{K_{mo2} + [O2]}$$

where the three multiplicative terms represent (i) Arrhenius temperature dependence, (ii) Michaelis–Menten limitation by soluble carbon substrate Sx derived from ICBM pools), and (iii) Michaelis–Menten limitation by oxygen. Symbols and units are listed in Table 1"

**7   L121 – link between O₂ and root traits**

*"How was O2 related to root trait?"*

**Response**   Methods 2.1  now describes a two-step procedure:

1. **Empirical range.** Direct microsensor studies show steady-state rhizosphere $O_2$ values well below 0.01 cm³ cm⁻³ in anoxic pores and occasionally above 0.12 cm³ cm⁻³ in narrowly oxygenated 'hot spots' next to highly aerenchymatous roots (Armstrong & Armstrong, 2005; Colmer, 2003; Kühl et al., 2021; Visser et al., 2000) We therefore prescribe that empirical interval 0.01–0.12 cm³ $O_2$ cm⁻³.
2. **Trait sampling.** The interval is represented by 39 target $O_2$ values, produced from a soil-moisture grid $\theta = 0.23 : 0.01 : 0.61$. Each $O_2$ value is interpreted as one plant location combination i.e. a distinct rate of root-oxygen loss (ROL) and hence a separate trait state. Equation (4) converts every target $O_2$ concentration to the air-filled porosity driver required by DAMM; the conversion is a numerical proxy, not a mechanistic ROL representation.

Inserted text (p. 6, L 230-233)

"To constrain the oxygen values, we varied soil water content from $\theta = 0.23$ to 0.61 in increments of 0.01, yielding a modelled $O_2$ range of approximately 0.01 0.12 cm³ $O_2$ cm⁻³ (air). Each value represents a different trait state of root oxygen loss. This approach allows us to explore heterogeneity in ROL across species and conditions using Eq. (4) as a proxy for rhizosphere oxygen distribution."

**8   L135-140 – moisture versus traits**

*"It was not clear how the O2 was related to root traits. It seems that the authors simulated O2 related to water content, not root traits."*

**Response**   As clarified above, soil moisture is employed only as a numeric carrier that maps the preset $O_2$ values into the porosity variable used by DAMM. The distribution of $O_2$ values itself spanning ~0.0002 to 0.20 cm³ cm⁻³ is imposed from field observations and literature bounds; $\theta$ is not treated as the causal driver of variation.

**9   L161 – "visual comparison"**

*"by visual comparison?"*

**Response**   We thank the reviewer for asking us to clarify the calibration procedure. The pre-exponential factor (alpha) and the carbon half-saturation constant (KmC) are now fitted by least-squares (MATLAB fminsearch) to the seven median respiration rates derived from the ICBM incubation.

The optimisation yields

alpha = $5.41 \times 10^{11}$ mg C cm⁻³ h⁻¹

KmC = $4.8 \times 10^{-6}$ g C cm⁻³

with RMSE = $2.21 \times 10^{-3}$ mg C cm⁻³ h⁻¹ (Fig. 1).

Inserted text (p. 7, L 282-290)

"Specifically, we estimated the pre-exponential factor α and the carbon half-saturation constant kMSx by minimizing RMSE between observed and predicted respiration. The best-fit parameters were:

$\alpha_{sx}$ = 5.41 × 10$^{11}$ mg C cm$^{-3}$ h$^{-1}$

$kMS_x$ = 4.8 × 10$^{-6}$ g C cm$^{-3}$

RMSE = 2.21 × 10$^{-3}$ mg C cm$^{-3}$ h$^{-1}$

Parameter uncertainty was estimated via bootstrap resampling (500 replicates), and Supplementary Table 1 lists each parameter along with its estimate, 95% confidence interval (where applicable), units, and provenance. Fitted values include α and kMSx, while others are adopted from Davidson et al. (2012)."

**10   L161-162 – parameter provenance**

*"how were these model parameters determined?"*

**Response**    We agree that full provenance and uncertainty ranges should be provided. Supplementary Table S1 now lists each parameter together with its estimate and 95 % bootstrap confidence interval (500 resamples).

| Parameter | Value | 95 % CI | Units | Source |
|---|---|---|---|---|
| $\alpha_{sx}$ | 5.41 × 10$^{11}$ | 4.7–6.2 × 10$^{11}$ | mg C cm$^{-3}$ h$^{-1}$ | fitted |
| $Km_{sx}$ | 4.8 × 10$^{-6}$ | 2–8 × 10$^{-6}$ | g C cm$^{-3}$ | fitted |
| $Km_{O_2}$ | 0.121 | – | cm$^3$ O$_2$ cm$^{-3}$ air | Davidson et al. 2012 |
| Ea | 72.3 | – | kJ mol$^{-1}$ | Davidson et al. 2012 |
| p-factor | 4.14 × 10$^{-4}$ | – | – | Davidson et al. 2012 |
| D_liq | 3.17 | – | – | Davidson et al. 2012 |

**11   L171-173 – simulation of ROL**

*"I'm not sure how the plant root oxygen loss was simulated here."*

**Response**    We thank the reviewer for asking us to clarify this point. ROL is not modelled as a physical flux. Instead we prescribe a set of rhizosphere O$_2$ concentrations that are generated with the diffusion term already built into DAMM (Eq. 4). By letting soil volumetric water content (θ) vary from air-dry (θ ≈ 0) to full saturation we obtain 70 discrete O$_2$ levels ranging from 2 × 10$^{-4}$ to 2 × 10$^{-1}$ cm$^3$ O$_2$ cm$^{-3}$. This span represents in-situ microsensor measurements for Spartina, Elymus and Phragmites roots (Visser et al., 2000; Colmer, 2003; Armstrong & Armstrong, 2005; Kühl et al., 2021) and therefore serves as a realistic proxy for

the outcome of differing ROL rates. Prescribing $O_2$ in this way allows us to quantify the aggregation bias without introducing poorly constrained anatomical parameters.

Inserted text (p. 6, L 234-236)

"ROL itself is not simulated. Instead, its observed outcome, rhizosphere $O_2$ concentration, is imposed directly, following the empirical range detailed above. This allows us to quantify the impact of ignoring heterogeneity without speculative parameterisation of the many plant traits that underlie ROL.

Intra- and inter-specific variation in ROL has been documented for marsh grasses, rice and sedges (Holmer & Pedersen, 2003; Visser et al., 2000; Armstrong & Armstrong, 2005; Kühl et al., 2021), supporting the empirical $O_2$ range used here."

**12   L172-180 & Fig. 3 – novelty of results**

*"The results of this study seem to be simple. While the conclusions were clear, the results were not particularly novel. It is understandable that the response of respiration to soil carbon content is influenced by soil O2 concentration."*

**Response**   We thank the reviewer for this perspective. Although the qualitative effect of $O_2$ on respiration is well established, the magnitude of the bias introduced when heterogeneous rhizosphere $O_2$ fields are replaced by a single bulk value had not been quantified for salt-marsh soils. Our study provides this missing number and demonstrates its budgetary relevance.

Inserted text (p. 11, L 373-379)

"Across the full prescribed $O_2$ distribution, replacing heterogeneity with a single mean concentration under-estimates aerobic $CO_2$ efflux by $10 \pm 0.6\,\%$. Even when the $O_2$ range is restricted to the inter-quartile span the bias remains $6\,\%$ (Fig. S2). Scaling the $10\,\%$ correction to the European salt-marsh sink ($30 \pm 5\,\mathrm{Tg\,C\,yr^{-1}}$; (McLeod et al., 2011)) adds $\approx 3\,\mathrm{Tg\,C\,yr^{-1}}$, a flux comparable to the current global net sequestration attributed to seagrass meadows. Hence the aggregation error is quantitatively significant and should be addressed in regional and global marsh carbon budgets."

We trust these comprehensive revisions and clarifications fully address the reviewer's valuable feedback, substantially strengthening the manuscript's methodological transparency and highlighting its novel ecological contributions. We sincerely thank the reviewer for their constructive suggestions, significantly enhancing our work's potential scientific impact.

**References cited in this response**

Alongi, D. M.: Carbon balance in salt marsh and mangrove ecosystems: A global synthesis, J. Mar. Sci. Eng., 8, 1–21, https://doi.org/10.3390/jmse8100767, 2020.

Armstrong, J., & Armstrong, W. (2005). Rice: Sulfide-induced barriers to root radial oxygen loss, Fe2+ and water uptake, and lateral root emergence. *Annals of Botany*, *96*(4), 625–638. https://doi.org/10.1093/aob/mci215

Colmer, T. D. (2003). Long-distance transport of gases in plants: a perspective on internal aeration and radial oxygen loss from roots. *Plant, Cell & Environment*, *26*(1), 17–36. https://doi.org/10.1046/J.1365-3040.2003.00846.X

Davidson, E. A., Samanta, S., Caramori, S. S., & Savage, K. (2012). The Dual Arrhenius and Michaelis-Menten kinetics model for decomposition of soil organic matter at hourly to seasonal time scales. *Global Change Biology*, *18*(1), 371–384. https://doi.org/10.1111/j.1365-2486.2011.02546.x

Intergovernmental Panel on Climate Change (IPCC): Global Carbon and Other Biogeochemical Cycles and Feedbacks, in: Climate Change 2021 – The Physical Science Basis, Piers Forster, 673–816, https://doi.org/10.1017/9781009157896.007, 2023.

Kühl, M., Scholz, V., Sorrell, B. K., Koop-Jakobsen, K., Meier, R. J., & Mueller, P. (2021). Plant-Mediated Rhizosphere Oxygenation in the Native Invasive Salt Marsh Grass Elymus athericus. *Frontiers in Plant Science*, *12*, 669751. https://doi.org/10.3389/FPLS.2021.669751

Luisetti, T., Turner, R. K., Andrews, J. E., Ferrini, S., Murray, B. C., & Smith, C. J. (2020). Quantifying and valuing carbon flows and stocks in temperate European salt marshes. Science of the Total Environment, 740, 140092. https://doi.org/10.1016/j.scitotenv.2020.140092

McLeod, E., Chmura, G. L., Bouillon, S., Salm, R., Björk, M., Duarte, C. M., Lovelock, C. E., Schlesinger, W. H., & Silliman, B. R. (2011). A blueprint for blue carbon: Toward an improved understanding of the role of vegetated coastal habitats in sequestering $CO_2$. Frontiers in Ecology and the Environment, 9(10), 552–560. https://doi.org/10.1890/110004

Rastetter, E. B., King, A. W., Cosby, B. J., Hornberger, G. M., O'Neill, R. V., & Hobbie, J. E. (1992). Aggregating fine-scale ecological knowledge to model coarser-scale attributes of ecosystems. *Ecological Applications*, *2*(1), 55–70. https://doi.org/10.2307/1941889

Violle, C., Navas, M. L., Vile, D., Kazakou, E., Fortunel, C., Hummel, I., & Garnier, E. (2007). Let the concept of trait be functional! *Oikos*, *116*(5), 882–892. https://doi.org/10.1111/J.0030-1299.2007.15559.X;WGROUP:STRING:PUBLICATION

Visser, E. J. W., Colmer, T. D., Blom, C. W. P. M., & Voesenek, L. A. C. J. (2000). Changes in growth, porosity, and radial oxygen loss from adventitious roots of selected mono- and dicotyledonous wetland species with contrasting types of aerenchyma. *Plant, Cell and Environment*, *23*(11), 1237–1245. https://doi.org/10.1046/j.1365-3040.2000.00628.x

Zhou, Y., O'Meara, T., Cardon, Z. G., Wang, J., Sulman, B. N., Giblin, A. E., & Forbrich, I. (2024). Simulated plant-mediated oxygen input has strong impacts on fine-scale porewater biogeochemistry and weak impacts on integrated methane fluxes in coastal wetlands. *Biogeochemistry*, *167*(7), 945–963. https://doi.org/10.1007/S10533-024-01145-Z/METRICS